# A Novel PV Array Reconfiguration Algorithm Approach to Optimising Power Generation across Non-Uniformly Aged PV Arrays by Merely Repositioning

**Mohammed Alkahtani [1]**, **Zuyu Wu [1]**, **Colin Sokol Kuka [2]**, **Muflah S. Alahammad [3]** and **Kai Ni [4],\***

[1] Department of Electrical Engineering and Electronics, University of Liverpool, Liverpool L69 3GJ, UK; m.alkahtani@liverpool.ac.uk (M.A.); wzy@liverpool.ac.uk (Z.W.)

[2] Department of Electronic Engineering, University of York, Heslington York YO10 5EZ, UK; sk1759@york.ac.uk

[3] Aerospace, Canfield University, Bedford MK43 0AL, UK; m.s.alhammad@cranfield.ac.uk

[4] School of Electrical and Electronic Engineering, Huazhong University of Science and Technology, Wuhan 430074, China

\* Correspondence: KevinNi93@hotmail.com; Tel.: +86-130-6387-2074

**Abstract:** Photovoltaic (PV) module working conditions lack consistency and PV array power outputs fluctuate due to the non-uniform impact that aging has on various PV modules in a PV array. No assessment has been conducted on the energy potential of a non-uniform PV array, despite the fact that the maximum power point (MPP) can be tracked by global maximum power point tracking (GMPPT). Therefore, the present work undertakes such an assessment by devising an algorithm to optimise the PV array electrical structure as the PV modules undergo aging in a non-uniform way. To enable PV arrays with non-uniform aging to produce as much power as possible and to make maintenance more cost-effective, the work puts forward a novel approach for reconfiguring PV arrays, where the PV modules are repositioned by retaining the aged PV modules. By this approach, the selection of the best reconfiguration topology necessitates the information on the electrical parameters associated with the PV modules in an array. Furthermore, the non-uniform aging of the PV modules can engender an incompatibility effect, which can be diminished in the proposed algorithm through iterative sorting of the modules in a hierarchical pattern. To determine how effective the method is for PV arrays with non-uniform aging and of different sizes, such as $3 \times 4$, $5 \times 8$ and $7 \times 8$ arrays, computer simulation and analysis have been conducted, with findings indicating that, irrespective of dimensions, PV arrays with non-uniform aging can have improved power yield.

**Keywords:** solar photovoltaic; rearrangement; maximum power point tracking; non-uniform aging; reconfiguration

---

## 1. Introduction

The new emphasis on clean energy has led to a growing interest in photovoltaic (PV) power production. To afford competitiveness to this new method of generating power, it must be made more energy-efficient and cost-effective. Having numerous applications in producing and transporting power and in mobile appliances, PV systems are embraced ever more widely, and it is anticipated that, by 2020, renewable sources will satisfy 20% of European energy demand [1,2]. In this context, PV plant-related financial and maintenance issues call for more efficient solar energy conversion and prolonging the useful life of PV arrays [3].

Transient obstacles (e.g. shadow, dust, bird droppings) and irreversible deterioration (e.g. suboptimal performance, PV cell/diode malfunction) can affect PV arrays when PV systems are in use. The National Renewable Energy Laboratory (NREL) [4,5], reported that PV arrays aged at different paces, and PV modules deteriorated in line with Gaussian distribution, and the annual pace at which PV modules deteriorated was 0.5% [6]. Thus, to prolong the service life and enhance the power yield of PV plants, the strategies for increasing PV power production for aged PV arrays must be explored. Substitution of aging PV modules with new ones is highly expensive, it is better to improve the efficiency of aged PV modules instead of replacing by new modules [7].

The warrants the formulation of a novel strategy for restructuring PV modules that are dysfunctional or old. Optimally, such an approach should involve the straightforward rearrangement of PV modules to increase the power yield [3,8]. This kind of strategy based on the bucket effect that stems from the PV string maximum short-circuit current ($I_{SC}$). It would be advantageous to have some basic knowledge of the general aspects of how PV arrays are organised and how they function [9].

In a PV array, PV modules can be organised and linked in various ways, with particular applications and properties being associated with every configuration format. Figure 1 shows the fundamental formats [10,11], namely, series-parallel (SP), total-cross-tied (TCT), and bridge-link interconnection (BLI). SP involves the in-series connection of modules and in-parallel correlation of ensuing rows. TCT involves the in-parallel connection of modules and in-series correlation of the configurations. BLI involves the linking of ties over junction rows [12,13].

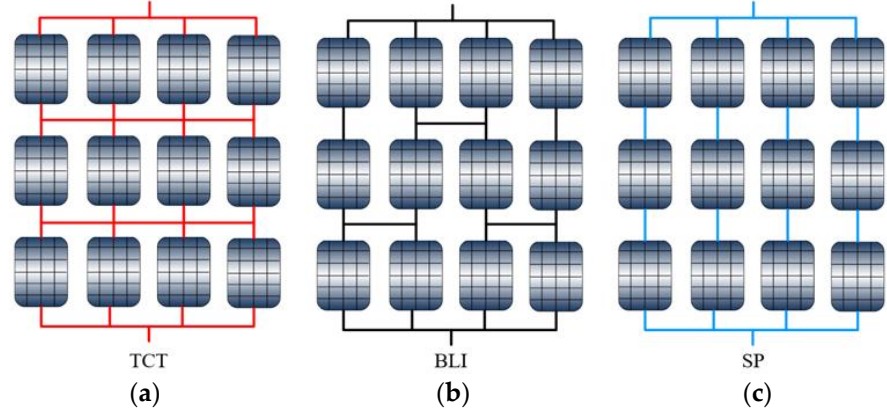

**Figure 1.** The fundamental formats of photovoltaic (PV) array linking are (**a**) total cross-tied (TCT), (**b**) bridge-link interconnection (BLI) and (**c**) series-parallel (SP).

The limitations of SP and BLI can be most effectively overcome based on TCT. TCT involves in-parallel correlation of PV modules so that each module has the same voltage and the current over a module row is additive, with the subsequent possibility of in-series connection of the rows of modules [14,15]. Studies investigating how different arrangements of PV arrays perform differ in terms of the arrangement formats they focus on, with some limiting themselves to fundamental series and parallel configurations, while others focus solely on TCT [16].

The multitude of options requiring consideration to establish the best solution is the main obstacle that has to be overcome for PV array rearrangement. Researchers have proposed different approaches in this regard. One approach proven to be suitable for sorting methods is to determine PV array rearrangement based on a genetic algorithm (GA) [17]. Meanwhile, other rearrangement approaches are geared towards enhancing power yield in settings with shade [18]. By prioritising the methods of array construction, however, [18] failed to implement real-time executable control algorithms, which resulted in an unfeasible number of sensors and switches requiring complicated control algorithms to detect on/off switch turning. Unlike the approach put forward in [18], a lower number of voltage or current sensors and switches are necessary for adaptive PV array rearrangement. In [19], an offline rearrangement approach was devised to make aged PV systems more energy efficient by inspecting

the possible options for PV module rearrangement based on the identification of the maximum power point. Meanwhile, in [1] the ideal arrangement for balancing and attenuating the aging process of switches in the switching matrix was assessed on the basis of the Munkres algorithm [20,21]. Issues related to the restructuring of modules in PV arrays of different sizes can be managed via additional approaches proven to be efficient, although these are computationally too complex and time-consuming because they involve a search of every possible manner, in which restructuring can be achieved [22].

This work primarily aims to propose an approach to repositioning PV modules to improve flaws or effects of aging of PV systems, thus increasing the power that a PV array can generate. In this context, speeding up the process of identifying the best arrangement is a key condition for the suggested algorithm. The structure of the rest of the work outlined below: Section 2 is the problem statement. Sections 3 and 4 describe the developed reconfiguration scheme for a non-uniformly aged PV array. Section 5 shows the simulation outcomes resulting from 4 × 3 and 8 × 5, 8 × 7 PV arrays. Section 6 presents restriction of inverter voltage. Section 7 presents a discussion of these findings. Section 8 reports the conclusions of our results and identifies the recommendations for future work.

## 2. Problem Statement

In the current part, the topology reconfiguration system presented, the physics underpinning PV array functioning is explained, the range of topologies applied practically, and associated electrical properties discussed, and the frameworks available for anticipating PV module electrical attributes presented.

This section may be divided by subheadings. It should provide a concise and precise description of the experimental results, their interpretation as well as the experimental conclusions that can be drawn.

### 2.1. Electrical Characteristics of A PV Cell

The current versus voltage (*I-V*) curve is the standard for representing the PV cell electrical features. The *I-V* attribute of a PV module illustrated in Figure 2. The short-circuit current is the superior left part of the *I-V* curve at zero voltage, while the open-circuit voltage is the inferior right part of the curve at zero current, and their measurements are derived respectively with the output terminals shorted (zero voltage) and the output terminals open (zero current).

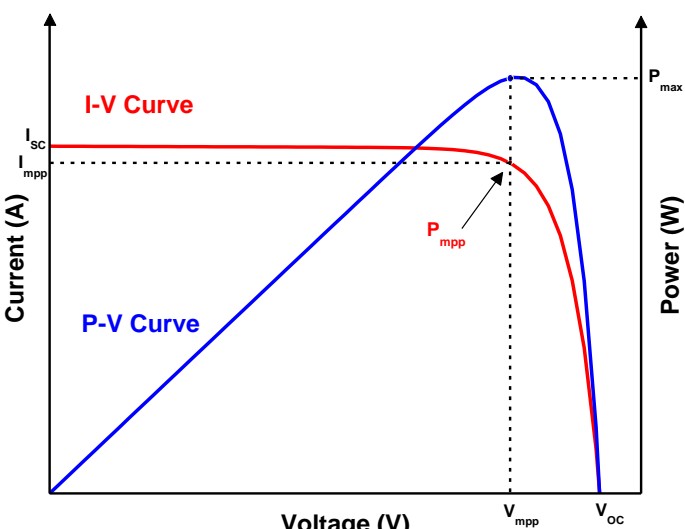

**Figure 2.** PV cell curve characteristics.

The voltage produced by the cell functioning as a source of constant current in the left part is equal to the load resistance. Meanwhile, a slight increase in voltage causes a fast decrease in the current in

the right part, where the cell functions as a source of constant voltage with internal resistance. The knee point of the curve is between these two parts [23].

## 2.2. System Description of PV Cell and PV Array

### 2.2.1. PV Cell Module

The purpose of equivalent circuit models is to represent the whole *I-V* curve of a cell/module/array in the form of a continuous function for a particular series of functioning circuit variables. Figure 3 illustrates the corresponding of a single diode circuit of the PV cell [24,25].

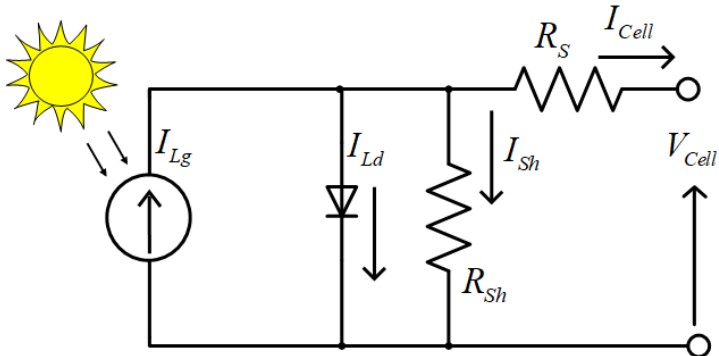

**Figure 3.** The equivalent of a single diode circuit of PV cell.

The equation for the equivalent circuit of the PV cell is formulated by using Kirchhoff's law for current $I_{Cell}$.

$$I_{Cell} = I_{Lg} - I_{Ld} - I_{Sh} \tag{1}$$

where, $I_{Lg}$ is the light-generated current in the cell, $I_{Ld}$ is loss diode-current and $I_{Sh}$ is the shunt-leakage current. In a single diode module, $I_{Ld}$ modeled using the Shockley equation for an ideal diode.

$$I_{Ld} = I_S \left[ \exp\left( \frac{(V_{Cell} + I_{Cell}R_S)q}{nV_T} \right) - 1 \right] \tag{2}$$

where the diode ideality factor between 1 and 2 for a single-junction cell is $n$, $I_s$ is the reverse saturation current of the diode, $V_{Cell}$ is the output voltage of the cell and $V_T$ is the voltage thermal can be expressed as:

$$V_T = \frac{kT_C}{q} \tag{3}$$

where the temperature of the *p-n* junction is $T_C$, $k$ is Boltzmann constant $1.38 \times 10^{-23} J/K$ and $q$ is the elementary charge $1.6 \times 10^{-19}C$.

Here, $R_S$ is the series resistance and $R_{Sh}$ the shunt is current can be xpressed as:

$$I_{Sh} = \frac{(V_{Cell} + I_{Cell}R_S)q}{R_{Sh}} \tag{4}$$

Then, the final single diode model can be expressed from the above equations results as:

$$I_{Ld} = I_{Lg} - I_S \left[ \exp\left( \frac{(V_{Cell} + I_{Cell}R_S)q}{nV_T} \right) - 1 \right] - \left( \frac{V_{Cell} + I_{Cell}R_S}{R_{Sh}} \right) \tag{5}$$

where the reverse saturation $I_S$ is current of the diode, $n$ is the diode factor and $V_T$ is the voltage thermal as expressed in Equation (3).

### 2.2.2. PV Array or Module

A cluster of multiple PV modules with in-series electrical linking and parallel circuits for producing the necessary current and voltage constitutes a PV array. Figure 4 presents the equivalent circuit associated with the PV module with $N_P$ parallel and $N_S$ series configuration [24].

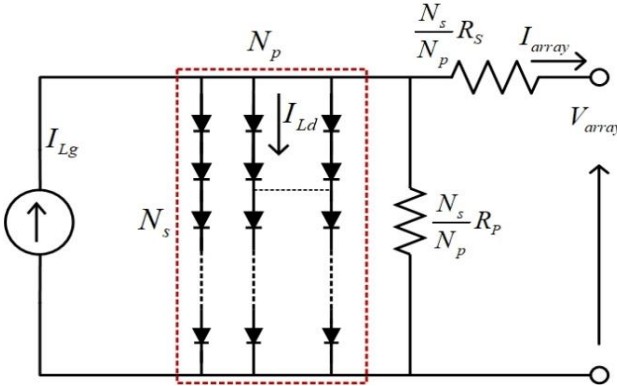

**Figure 4.** Equivalent circuit of PV array.

Therefore, for the PV array, as shown in Figure 4, the output current equation as given below:

$$I_{array} = N_p I_{Lg} - N_p I_{Ld} I_S \left[ \exp\left( \frac{\left( V_{array} + I_{array} \frac{N_s}{N_p} R_S \right) q}{n V_T N_s} \right) - 1 \right] - \frac{V_{array} + I_{array} \frac{N_s}{N_p} R_S}{\frac{N_s}{N_p} R_{Sh}}. \tag{6}$$

For a PV array containing $N_s$ cells in series and such $N_p$ strings in parallel, $V_{array}$ is the bandgap voltage, $I_{array}$ is the current, respectively as shown in Figure 4. Therefore, the reverse saturation $I_S$ is current of the diode, $n$ is the diode factor and $V_T$ is the voltage thermal as expressed in Equation (3).

Simulation and representation were based on the Solarex (MSX60) PV module [26] comprising 36 polycrystalline cells with in-series linking (Table 1).

**Table 1.** The disclaimers of the Solarex (MSX60) photovoltaic module [26].

| PV Panel Parameters | Symbols | Values |
|---|---|---|
| Open-circuit voltage | $V_{OC}$ | 21.1 V |
| Short-circuit current | $I_{SC}$ | 3.8 A |
| Maximum power | $P_{max}$ | 60 W |
| Maximum power current | $I_{mp}$ | 3.5 A |
| Maximum power voltage | $V_{mp}$ | 17.1 V |
| Cell-temperature | $T$ | 25 °C |

Information related to an open-circuit voltage ($V_{OC}$), short-circuit current ($I_{SC}$) and maximum power point ($V_{mp}$, $I_{mp}$) at standard test conditions (STC) is typically specified by MSX60 PV producers [22,27].

- Short-circuit current ($I_{SC}$) is the maximum current that a PV module can generate.
- Open-circuit voltage ($V_{OC}$) is the maximum voltage across a PV module.
- Maximum power point (MPP) is the point on the *I-V* (voltage-current) characteristic curve where the product of voltage $V_{mp}$ and current $I_{mp}$, is the maximum.

### 2.3. Analysis of the Mismatch Due to the Non-Uniformly Aging

The short circuit current differs more than the open-circuit voltage when a PV cell meet with aging experiment, because of the *p-n* junction qualities of the cell according to the authors of [7,22]. This work evaluates the PV module aging status based on the short-circuit current while maintaining the open-circuit voltage unchanged for different aging conditions. Additionally, it is assumed that every one of the cell-units in the same PV module experiences uniform aging, thus that whole PV module may be characterised with a single maximum short-circuit of any of the cell-units. In the case of *m* PV modules with in-series connection making up a PV array, their output currents will be the same, while the total module voltages will be added up to obtain the output voltage [7].

$$I_{total} = I_{module(1)} = I_{module(2)} = I_{module(3)} = \cdots = I_{module(m)} \tag{7}$$

$$V_{total} = \sum_{\tau=1}^{m} V_{module(\tau)} = V_{module(1)} + V_{module(2)} + V_{module(3)} + \cdots + V_{module(m)} \tag{8}$$

As shown in Figure 5, in the best scenario, the behavior of the modules does not differ, and the voltages are equal to the value of 63 V for all the three modules. The same, the short-circuit current going through PV modules within series-connection is equal to 3.8 A for each.

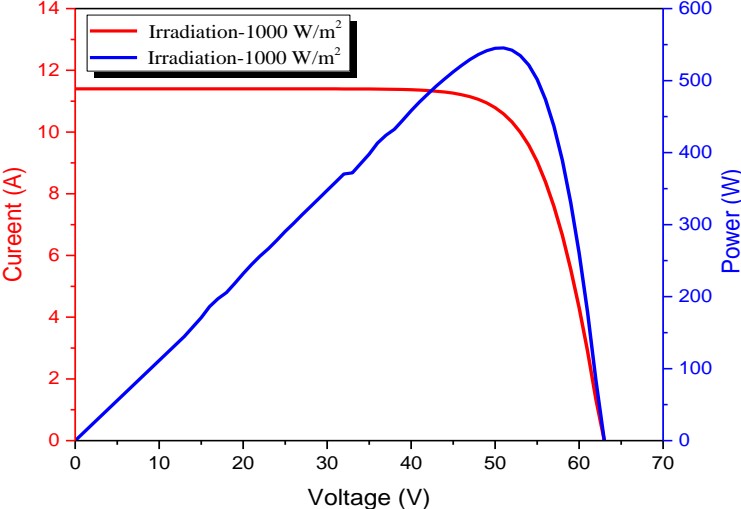

**Figure 5.** Simulation responses: *I-V* curve and *P-V* curve (at standard test conditions) for good quality modules of Solarex MSX60 connected in series.

The short-circuit current of separate PV modules may become incompatible in the context of non-uniform aging. To avoid hot spots, all PV modules have in parallel-connecation to a bypass diode. Three modules connected in series with aging to various degrees. Where the maximum short-circuit current of each PV module is given in per unit (pu) and then the aging condition can be expressed as 0.9 pu; 0.6, 0.5 pu as shown in Figure 6.

Due to the non-uniform aging conditions, multiple PV power output steps and peaks observed in Figure 7 divides the PV array operation into three different operational levels, where each peak relates to a particular level. Level 1 indicates a phase where module 1 is active, while the currents across modules 2 and 3 are being bypassed through the diodes. At level 1, the current ranges between 0 A and 3.45 A, and the corresponding voltage is 0–22 V. Similarly, level 2 corresponds to the phase where module 1 and module 2 are active, while module 3 is being bypassed.

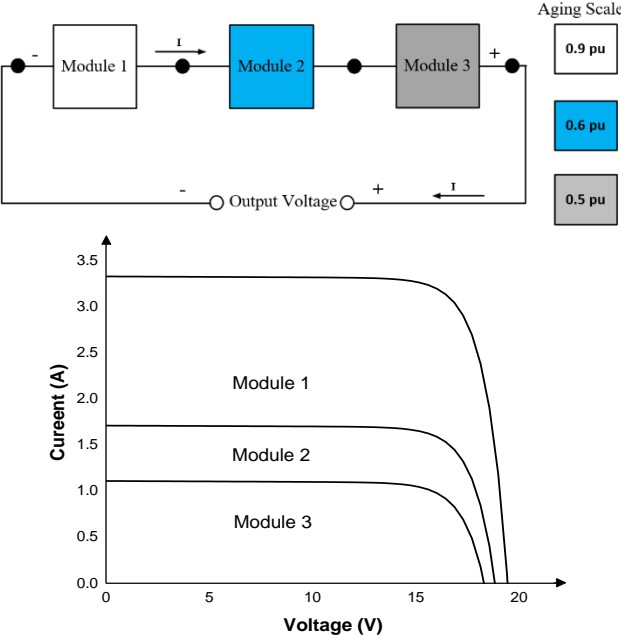

**Figure 6.** Series-connection of PV modules.

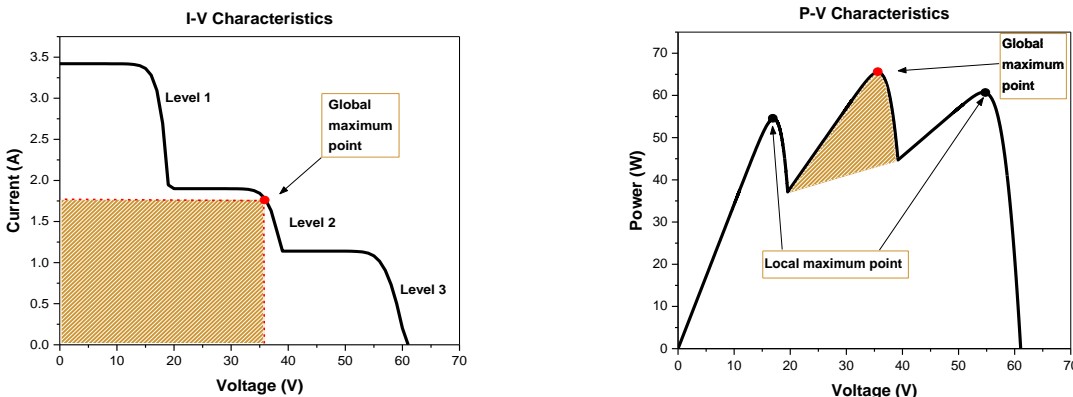

**Figure 7.** Series-connection of PV modules.

The current for the un-bypassed series-connection is determined by the current of the most aged PV module (in this case, module 2). The level 2 current ranges between 0 A and 2.30 A, with a corresponding voltage of 0–42 V. Level 3 represents the stage where modules are active, that is, none are bypassed. Again here, the current for the un-bypassed series connection is determined by the current of the most aged PV module (i.e., module 3). The current at level 3 ranges between 0 A and 1.5 A, with a corresponding voltage of 0–82 V. Moreover, there are many maximum power points expressed by the knee points for the various levels in the characteristic *I–V* curve. These knee points are correlated with particular currents and voltages that are utilised to derive the maximum power points at various locations on the P–V curve. The knee point at level 1 arrives at 16.79 V and 3.45 A (54.42 W); the knee point at level 2 is at 35.69 V, and 2.30 A (65.45 W); whilst the knee point at level 3 is at 54.9 V and 1.5 A (60.61 W), as depicted in Figure 6 the knee point at level 2 indicates the global maximum power point (GMPP). The knee point at level 1 tasks at 16.79 V and 3.45 A (54.42 W); The knee point at level 2 tasks at 35.69 V and 2.30 A (65.45 W); whilst the knee point at level 3 tasks at 54.9 V and 1.5 A (60.61 W), as depicted in Figure 7 the knee point at level 2 indicates the global maximum power point (GMPP).

Furthermore, for PV aging map measurement, the most popular way is to use time-domain reflectometry (TDR) incompatible PV array of superior performance being obstructed by modules of

inferior performance and overheated linkages, yet having identical nominal power, can be detected via time-domain reflectometry TDR. This is also a suitable method for identifying dysfunctional PV modules while maintaining module connection [28]. Consequently, by the TDR, the PV array is detected in the night, because the TDR only can be used in the non-illumination condition. Furthermore, in the night, it is safe for the electrical engineers to do the corresponding testing work. Therefore, it will not happen to stop the PV generation system in electric generation condition.

In the following section, the series-connected modules might structure strings, which need assistance joined in parallel to structure an SP configuration for non-uniform conditions, therefore on illustrate our proposed reconfiguration algorithmic rule.

## 3. PV Array Reconfiguration Scheme

In an $N \times M$ PV array as shown in Figure 8. Where is $N$ is the parallel linking and $M$ is the series linking PV modules. The voltage at which the PV array GMPP is found in the P–V curve gives the number of active modules for a particular string voltage. Therefore, by adding up all the string currents and multiplying that figure by the string voltage of the active modules, the PV array maximum power can be obtained.

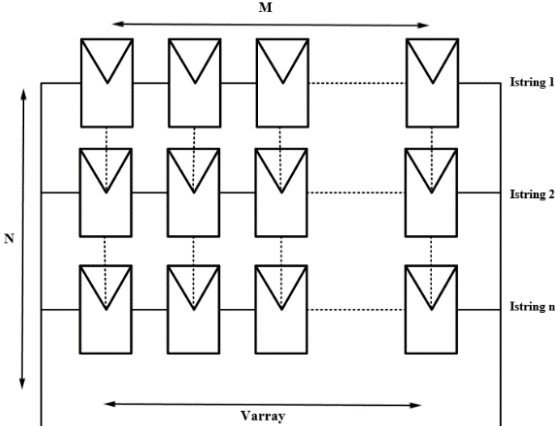

**Figure 8.** A series-parallel (SP) PV array involving $N \times M$ (number of parallel-connected strings × number of series-connected PV modules).

A PV array comprising 12 aged modules connected in a $3 \times 4$ PV array SP configuration (as depicted in Figure 9) will be utilised to demonstrate this concept.

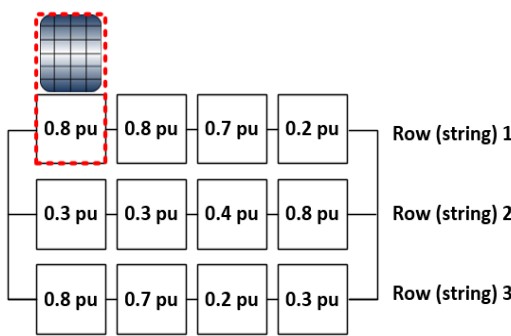

**Figure 9.** A $3 \times 4$ PV array SP configuration with non-uniform aging.

## 4. Reconfiguration Algorithm of PV Array

An $N \times M$ PV array can typically have $\left(\frac{NM}{M}\right)\left(\frac{(N-1)M}{M}\right)\left(\frac{(N-2)M}{M}\right)\ldots\left(\frac{2M}{M}\right)\left(\frac{M}{M}\right)/N!$ arrangements. This means that the number of potential methods for a $3 \times 4$ PV array will be 1,227,656, which will make it

extremely challenging to determine the maximum power for every possible PV module arrangement for a PV array with larger N and M values. To attain the best configuration in a few iterative steps, a new reconfiguration algorithm is put forth, drawing on the principle of sorting PV modules repetitively and hierarchically. Given its close correlation with the short-circuit current of all PV modules, the aging scale (coefficient) serves as the varying parameter in the suggested algorithm. Five modules representing different levels of solar irradiance:

Module 1: solar irradiance 200 W/m$^2$ and temperature 25 °C.
Module 2: solar irradiance 400 W/m$^2$ and temperature 25 °C.
Module 3: solar irradiance 600 W/m$^2$ and temperature 25 °C.
Module 4: solar irradiance 800 W/m$^2$ and temperature 25 °C.
Module 5: solar irradiance 1000 W/m$^2$ and temperature 25 °C.

In a suitable module, the STC specifies the short-circuit current to be 1 per unit (pu), which corresponds to 1000 W/m$^2$. The digits indicate the various aging factors (AF) associated with the PV modules in the array, is directly correlated with their separate short-circuit current. For instance, the optimisation issue addressed in the present work is based on an iterative and hierarchical sorting algorithm, called selection sort and use for iteration steps to achieve optimum configuration, which applied to a PV array arrangement Figure 9. The AFs take the form of pu value of the health condition of separate PV modules and represent the working box variables. The rules suggested for this work are listed below.

- The first rule specifies that equivalence exists between string one working box, string two working box and string $n$ working box. Means that both string two and string $n$ will have three working boxes if the string is associated with three working boxes.
- The second rule specifies that, in a string, the minimal number represent the working box output. Its means that the output is the lowest among all values from high to low.
- $P_{string(n)} = \sum AF$ = Summation of aging factors in a series of connected modules.

A. Pre-arrangement can be mathematically characterised within five steps.

**Step 1**: Initialize the summation of AFs for each string Pre-arrangement, as follows:

| | | | |
|---|---|---|---|
| 0.8 pu | 0.8 pu | 0.7 pu | 0.2 pu |
| 0.3 pu | 0.3 pu | 0.4 pu | 0.8 pu |
| 0.8 pu | 0.7 pu | 0.2 pu | 0.3 pu |

Sum:

$$P_{string1} = 0.8 + 0.8 + 0.7 + 0.2 = 2.5$$
$$P_{string2} = 0.3 + 0.3 + 0.4 + 0.8 = 1.8 \qquad (9)$$
$$P_{string3} = 0.8 + 0.7 + 0.2 + 0.3 = 2$$

**Step 2:** Arrange the working boxes of $P_{4(total)}$ Pre-arrangement in descending order, in the case study.

Select lowest number:

| 0.8 pu | 0.8 pu | 0.7 pu | 0.2 pu | Select lowest number of $P_4$ string n |
|---|---|---|---|---|
| 0.3 pu | 0.3 pu | 0.4 pu | 0.8 pu | |
| 0.8 pu | 0.7 pu | 0.2 pu | 0.3 pu | |

$$P_{string1} = 0.2 \times 4 = 0.8$$
$$P_{string2} = 0.3 \times 4 = 1.2 \tag{10}$$
$$P_{string3} = 0.2 \times 4 = 0.8$$

Sum:

$$P_{4(total)} = P_{string1} + P_{string2} + P_{string3} = 2.8 \tag{11}$$

**Step 3:** Arrange the working boxes of $P_{3(total)}$ Pre-arrangement in descending order, in the case study.

| | | | |
|---|---|---|---|
| 0.8 pu | 0.8 pu | 0.7 pu | 0.2 pu |
| 0.3 pu | 0.3 pu | 0.4 pu | 0.8 pu |
| 0.8 pu | 0.7 pu | 0.2 pu | 0.3 pu |

$$P_{string1} = 0.7 \times 3 = 2.1$$
$$P_{string2} = 0.3 \times 3 = 0.9 \tag{12}$$
$$P_{string3} = 0.2 \times 3 = 0.6$$

Sum:

$$P_{3(total)} = P_{string1} + P_{string2} + P_{string3} = 3.6 \tag{13}$$

**Step 4:** Arrange the working boxes of $P_{2(total)}$ Pre-arrangement in descending order.

| | | | |
|---|---|---|---|
| 0.8 pu | 0.8 pu | 0.7 pu | 0.2 pu |
| 0.3 pu | 0.3 pu | 0.4 pu | 0.8 pu |
| 0.8 pu | 0.7 pu | 0.2 pu | 0.3 pu |

$$P_{string1} = 0.8 \times 2 = 1.6$$
$$P_{string2} = 0.4 \times 2 = 0.8 \tag{14}$$
$$P_{string3} = 0.7 \times 2 = 1.4$$

Sum:

$$P_{2(total)} = P_{string1} + P_{string2} + P_{string3} = 3.8. \tag{15}$$

**Step 5:** Arrange the working box of $P_{1(total)}$ Pre-arrangement in descending order.

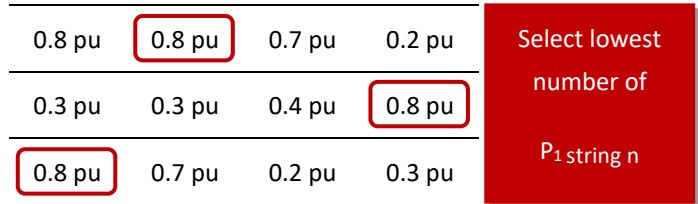

$$P_{string1} = 0.8$$
$$P_{string2} = 0.8 \tag{16}$$
$$P_{string3} = 0.8$$

Sum:

$$P_{1(total)} = P_{string1} + P_{string2} + P_{string3} = 2.4 \tag{17}$$

B. The potential PV array arrangements from initial to final string must be identified sequentially.

As shown by the equation below, the PV array takes the form of a matrix to facilitate the running of the MATLAB program.

$$N \times M = \begin{bmatrix} 0.8 & 0.8 & 0.7 & 0.2 \\ 0.3 & 0.3 & 0.4 & 0.8 \\ 0.8 & 0.7 & 0.2 & 0.3 \end{bmatrix} \tag{18}$$

Figure 10 illustrates the flowchart associated $N \times M$ with the rearrangement algorithm for the PV array. The suggested algorithm geared towards mitigating the impact of mismatch losses between the PV modules in a given string by relocating separate PV modules in every string according to their AFs. Due to the direct correlation between aging and the short-circuit current, AFs are the only short-circuit current data needed by the algorithm. To attain the best arrangement, the algorithm run until every criterion is satisfied Figure 10.

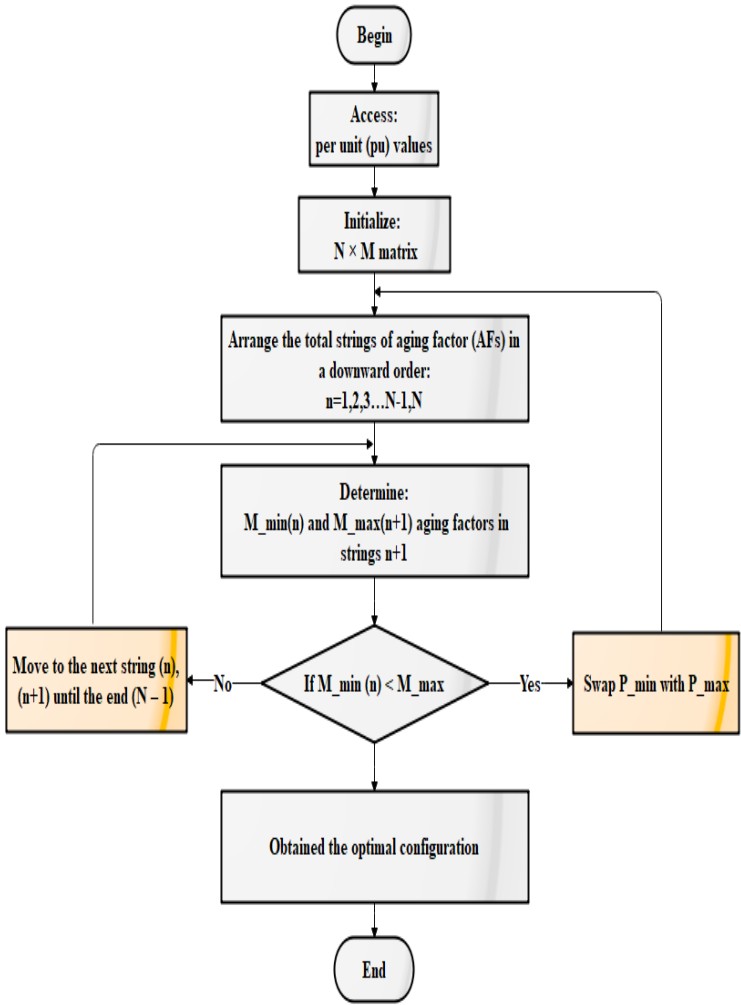

**Figure 10.** Flowchart of PV reconfiguration algorithm process.

Before presenting the five steps of the suggested algorithm, several parameters need to be described to elucidate the rearrangement approach from the previous flowchart.

$n = 1, 2, 3 \ldots N - 1, N$, where the number of strings in the PV array called $N$.

- $\sum AF$ = Summation of aging factors in a series of connected modules.
- $M_{string(min)n}$ = Minimum AFs in a series connection for a string $(n)$.
- $M_{string(max)n}$ = Maximum AFs in a series connection for a string $(n + 1)$.
- $P_{string(min)}$ = Position of PV module with a minimum AFs in a series of connected modules.

- $P_{string(\max)n+1}$ = Position of PV module with a maximum AFs in a series of connected modules.

**Step 1:** Initialize the summation of $P_{string(n)} \approx$ AFs for each string and arrange the total string level $P_{string(n)}$ in descending order, in the case study.

$$P_{string1} = 0.8 + 0.8 + 0.7 + 0.2 = 2.5$$
$$P_{string2} = 0.3 + 0.3 + 0.4 + 0.8 = 1.8 \tag{19}$$
$$P_{string3} = 0.8 + 0.7 + 0.2 + 0.3 = 2$$

**Step 2:** Arrange the total string level AFs in a downward order in the case study.

**Step 3:** Determine $M_{\min(n)}$ and $M_{\max(n+1)}$ for $n = 1$

| 0.8 pu | 0.8 pu | 0.7 pu | 0.2 pu |
| 0.3 pu | 0.3 pu | 0.4 pu | 0.8 pu |
| 0.8 pu | 0.7 pu | 0.2 pu | 0.3 pu |

$M_{\min A}$

$M_{\max B}$

Now, if $M_{\min A} < M_{\max B}$, then swap $P_{\min}$ with $P_{\max}$, repeat steps 1, 2 and 3.

| 0.8 pu | 0.8 pu | 0.7 pu | 0.8 pu |
| 0.3 pu | 0.3 pu | 0.4 pu | 0.2 pu |
| 0.8 pu | 0.7 pu | 0.2 pu | 0.3 pu |

Swap₁

| 0.8 pu | 0.8 pu | 0.7 pu | 0.8 pu |
| 0.8 pu | 0.7 pu | 0.2 pu | 0.3 pu |
| 0.3 pu | 0.3 pu | 0.4 pu | 0.2 pu |

Swap₂

Then, $P_{string(n)}$ on the left-hand side are:

$$P_{string1} = 0.8 + 0.8 + 0.7 + 0.8 = 3.1$$
$$P_{string2} = 0.3 + 0.3 + 0.4 + 0.2 = 1.2 \tag{20}$$
$$P_{string3} = 0.8 + 0.7 + 0.2 + 0.3 = 2$$

Moreover, $P_{string(n)}$ on the right-hand side are:

$$P_{string1} = 0.8 + 0.8 + 0.7 + 0.8 = 3.1$$
$$P_{string2} = 0.8 + 0.7 + 0.2 + 0.3 = 2 \tag{21}$$
$$P_{string3} = 0.3 + 0.3 + 0.4 + 0.2 = 1.2$$

**Step 4:** Repeat steps 1, 2 and 3 till $M_{\min(n)} \geq M_{\max(n+1)}$.

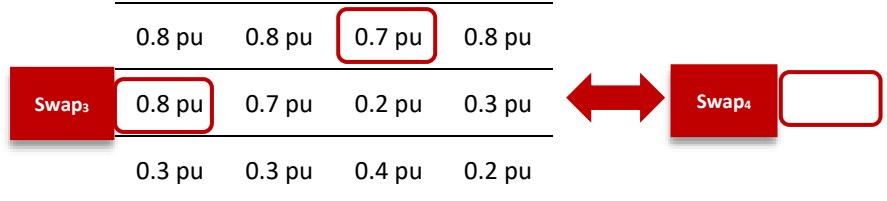

| | | | |
|---|---|---|---|
| 0.8 pu | 0.8 pu | 0.8 pu | 0.8 pu |
| 0.7 pu | 0.7 pu | 0.2 pu | 0.3 pu |
| 0.3 pu | 0.3 pu | 0.4 pu | 0.2 pu |

Then, the sum of $P_{string(n)}$ the left-hand side are:

$$
\begin{aligned}
P_{string1} &= 0.8 + 0.8 + 0.7 + 0.8 = 3.1 \\
P_{string2} &= 0.8 + 0.7 + 0.2 + 0.3 = 2 \\
P_{string3} &= 0.3 + 0.3 + 0.4 + 0.2 = 1.2
\end{aligned}
\tag{22}
$$

Moreover, on the right-hand side are:

$$
\begin{aligned}
P_{string1} &= 0.8 + 0.8 + 0.8 + 0.8 = 3.2 \\
P_{string2} &= 0.7 + 0.7 + 0.2 + 0.3 = 1.9 \\
P_{string3} &= 0.3 + 0.3 + 0.4 + 0.2 = 1.2
\end{aligned}
\tag{23}
$$

**Step 5:** Find $M_{\min(n)}$ and $M_{\max(n+1)}$ for $n = 2$, swap the corresponding $P_{\min}$ with and $P_{\max}$ repeat steps 1 and 2 until the end (N–1).

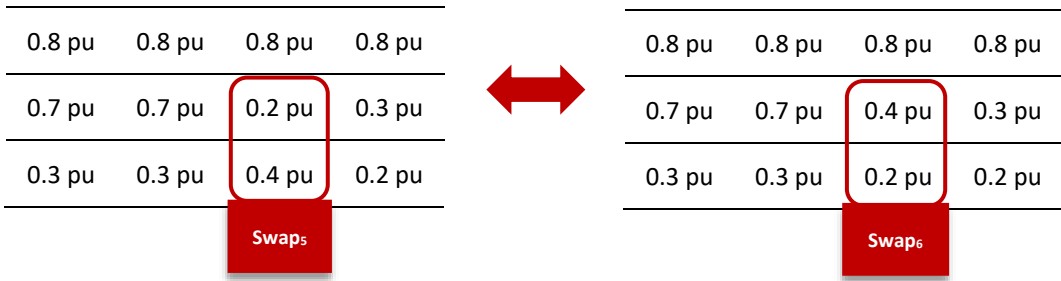

Then, the final step, the sum of $P_{string(n)}$ in the left-hand side are:

$$
\begin{aligned}
P_{string1} &= 0.8 + 0.8 + 0.8 + 0.8 = 3.2 \\
P_{string2} &= 0.7 + 0.7 + 0.2 + 0.3 = 1.9 \\
P_{string3} &= 0.3 + 0.3 + 0.4 + 0.2 = 1.2
\end{aligned}
\tag{24}
$$

In the right-hand side are:

$$
\begin{aligned}
P_{string1} &= 0.8 + 0.8 + 0.8 + 0.8 = 3.2 \\
P_{string2} &= 0.7 + 0.7 + 0.4 + 0.3 = 2.1 \\
P_{string3} &= 0.3 + 0.3 + 0.2 + 0.2 = 1
\end{aligned}
\tag{25}
$$

According to the final step, the best arrangement exhibited by the PV array on the right-hand side (RHS). Nevertheless, a comparison conducted between every arrangement arriving at every step and the initial arrangement Figure 9. Under non-uniform aging conditions, the ideal arrangement was obtained solely through five repetitive steps for a 3 × 4 PV array. In the case of a large PV array, execution based on a MATLAB program, with the configuration for the best power yield being represented by the enhanced form. Hence, the ideal arrangement for a 3 × 4 PV array is the PV array Post-arrangement. The PV arrays of Pre-Post arrangements are compared in Table 2.

**Table 2.** PV Array Pre-Post rearrangements.

| Pre-Arrangement | | | |
|---|---|---|---|
| 0.8 pu | 0.8 pu | 0.7 pu | 0.2 pu |
| 0.3 pu | 0.3 pu | 0.4 pu | 0.2 pu |
| 0.8 pu | 0.7 pu | 0.2 pu | 0.3 pu |
| Post-Arrangement | | | |
| 0.8 pu | 0.8 pu | 0.8 pu | 0.8 pu |
| 0.7 pu | 0.7 pu | 0.4 pu | 0.3 pu |
| 0.3 pu | 0.3 pu | 0.2 pu | 0.2 pu |

In Table 3, the maximum power and voltage at MPP are set out for all arrangements and the string currents in every case. It is obvious that, from the first to the fifth step, there is a 22.4% rise in the overall output power and the voltage at MPP is greater than the output current. To minimise multiple peaks caused by incompatibility effects (non-uniform aging), the proposed algorithm increases the currents in every string as much as possible through the integration of the PV modules showing similar electrical features.

**Table 3.** Electrical parameters obtained for different reconfiguration.

| Steps | $V_{mpp}$ (V) | Maximum $P_{mpp}$ (W) | String Current (A) | | |
|---|---|---|---|---|---|
| | | | $I_1$ | $I_2$ | $I_3$ |
| 1 | 53 | 254.3 | 2.531 | 1.141 | 1.143 |
| 2 | 70 | 286.9 | 2.587 | 0.759 | 0.751 |
| 3 | 70 | 287.1 | 2.586 | 0.758 | 0.751 |
| 4 | 69 | 297.7 | 2.798 | 0.761 | 0.756 |
| 5 | 68 | 320.8 | 2.844 | 1.142 | 0.728 |

## 5. Simulation Results

PV arrays of different dimensions (e.g. $3 \times 4$, $5 \times 8$ and $7 \times 8$) were assessed to prove that the suggested algorithm was valid. A MATLAB-developed PV array model was used to compute the maximum power outputs from the PV structures pre-arrangement as well as post-arrangement. The computations were conducted with an Intel® Core™ computer with i3-3220 CPU, 30.30 GHz and 8 GB (RAM), with tabulation of the equivalent computing times for the different PV array dimensions indicated above.

A.    Case study on $3 \times 4$ PV array

Figure 9 shows the MATLAB-based validation of the results. Under STC, the maximum short-circuits current in a suitable module established at 1 pu, which is equivalent to 1000 W/m$^2$ irradiance at a module temperature of 25 °C.

Table 2 shows the PV configuration following a rearrangement using the proposed algorithm. Using the PV array data presented in Table 2, *I-V* and *P-V* curves were then plotted as depicted in Figure 11. In Figure 11 highlights that the maximum output power pre-arrangement, is 247.4 W, with a PV array output voltage of 51 V and a GMPP current of 4.8 A, respectively. The maximum output power post-arrangement is 320.8 W, with a PV array output voltage of 68 V and GMPP current of 4.68 A, respectively. It can be seen that the total power output increases by 29.7% as presented in Figure 11 when using the proposed algorithm. The computational time for these rearrangements (as presented in Table 4) for an aged $3 \times 4$ PV array took 0.02 s.

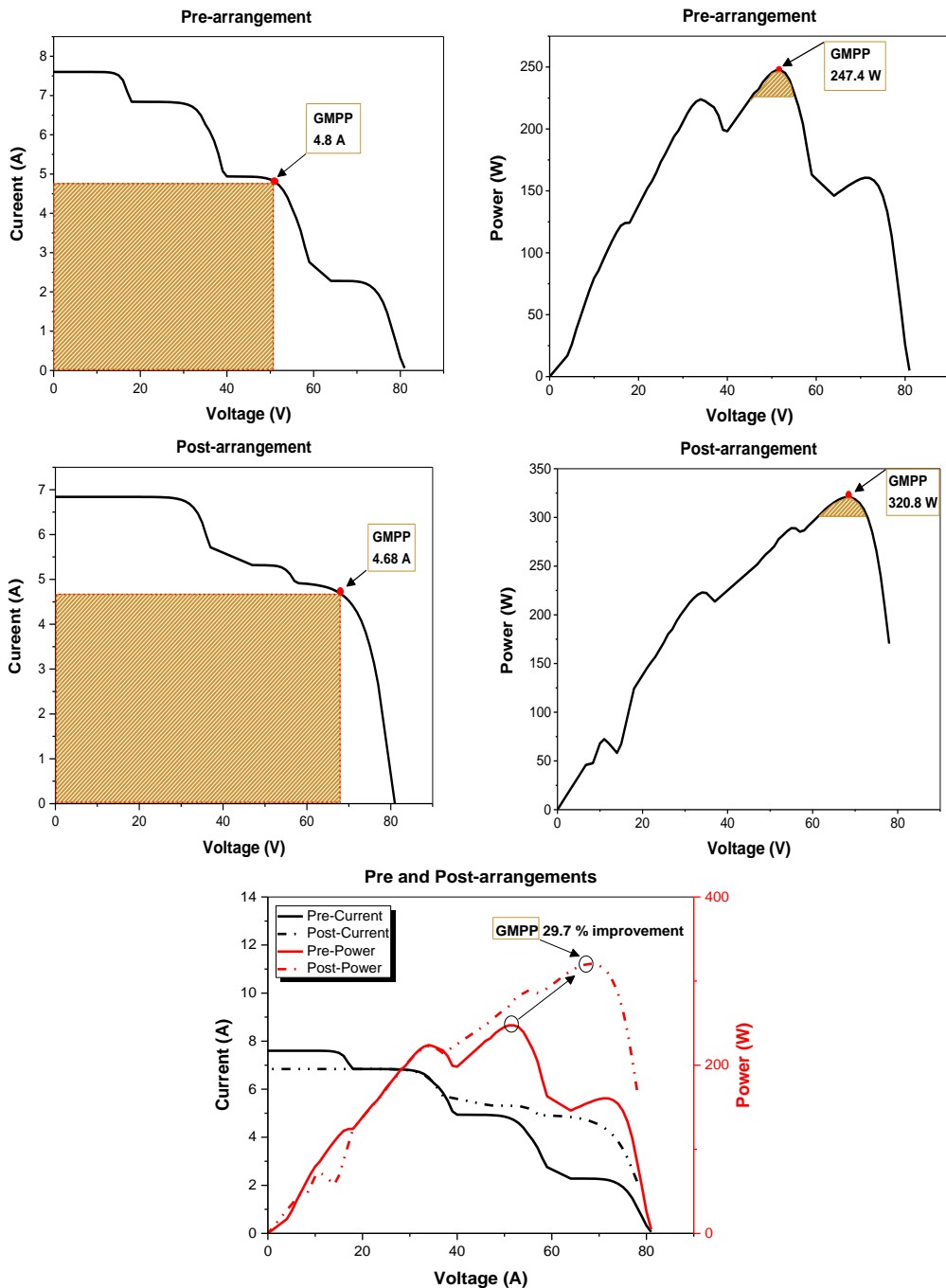

**Figure 11.** The output of the Array (pre-post rearrangements) for case 1.

**Table 4.** PV array 3 × 4 parameters of Pre-Post arrangements.

| PV Array 3 × 4 Parameters | | | | |
|---|---|---|---|---|
| **Parameters** | **Pre-Arrangement** | **Post-Arrangement** | **Power Improvement** | **Computing Time (s)** |
| Current $I_{mpp}$ | 4.8 A | 4.68 A | – | – |
| Voltage $V_{mpp}$ | 51 V | 68 V | 29.7% | 0.02 |
| Power $W_L$ | 247.4 W | 320.8 W | – | – |

B.    Case study on 5 × 8 PV array

The PV array of dimensions 5 × 8 consisted of five strings and eight modules with in parallel and in-series linking, respectively. For developing a 5 × 8 matrix, simulating non-uniform aging PV array pre-arrangement and determine the best PV structure post-arrangement for this particular case, MATLAB (R2018a) employed for arbitrary production of the AFs in the range 0.9-0.6 pu (Table 5). The ability of the suggested algorithm to yield the ideal arrangement was confirmed by simulating both PV structures. Figure 12 illustrates that the maximum power output pre-arrangement is 1722 W, with a PV array output voltage of 143 V and a GMPP current of 11.9 A, respectively. The maximum power output post-arrangement is 1885 W, with a PV array output voltage of 138 V and a GMPP current of 13.6 A, respectively. The computational time for the proposed algorithm to identify the rearrangements of an aged 8 × 5 PV array (as presented in Table 6) took 0.25 s.

**Table 5.** PV array configuration for case 2.

| Pre-arrangement | | | | | | | |
|---|---|---|---|---|---|---|---|
| 0.9 pu | 0.8 pu | 0.9 pu | 0.9 pu | 0.8 pu | 0.9 pu | 0.9 pu | 0.7 pu |
| 0.8 pu | 0.9 pu | 0.7 pu | 0.8 pu | 0.9 pu | 0.9 pu | 0.9 pu | 0.8 pu |
| 0.7 pu | 0.9 pu | 0.8 pu | 0.9 pu | 0.8 pu | 0.7 pu | 0.6 pu | 0.7 pu |
| 0.8 pu | 0.8 pu | 0.9 pu | 0.7 pu | 0.7 pu | 0.6 pu | 0.7 pu | 0.6 pu |
| 0.9 pu | 0.7 pu | 0.8 pu | 0.9 pu | 0.9 pu | 0.8 pu | 0.8 pu | 0.6 pu |
| **Post-arrangement** | | | | | | | |
| 0.9 pu | 0.9 pu | 0.9 pu | 0.9 pu | 0.9 pu | 0.9 pu | 0.9 pu | 0.9 pu |
| 0.9 pu | 0.9 pu | 0.9 pu | 0.9 pu | 0.9 pu | 0.9 pu | 0.9 pu | 0.8 pu |
| 0.8 pu | 0.8 pu | 0.8 pu | 0.8 pu | 0.8 pu | 0.8 pu | 0.8 pu | 0.8 pu |
| 0.8 pu | 0.8 pu | 0.7 pu | 0.7 pu | 0.7 pu | 0.7 pu | 0.7 pu | 0.7 pu |
| 0.6 pu | 0.7 pu | 0.7 pu | 0.6 pu | 0.6 pu | 0.7 pu | 0.7 pu | 0.6 pu |

**Table 6.** PV array 5 × 8 parameters Pre and Post-arrangement.

| PV Array 5 × 8 Parameters | | | | |
|---|---|---|---|---|
| **Parameters** | **Pre-Arrangement** | **Post-Arrangement** | **Power Improvement** | **Computing Time (s)** |
| Current $I_{mpp}$ | 11.9 A | 13.6 A | – | – |
| Voltage $V_{mpp}$ | 143 V | 138 V | 9.47% | 0.25 |
| Power $W_L$ | 1722 W | 1885 W | – | – |

C.    Case study on 7 × 8 PV array

In this case, an 7 × 8 PV array was, comprising of seven parallel-connected strings and eight series-connected modules, The aging factors, ranging from 0.9 pu to 0.4 pu (as shown in Table 7), were randomly generated, as in case 1 and 2. Figure 13 illustrates that the total output power increases by 32.5% when the proposed algorithm used. The computational time for the proposed algorithm to identify the rearrangements (as presented in Table 8) took time 5.64 s.

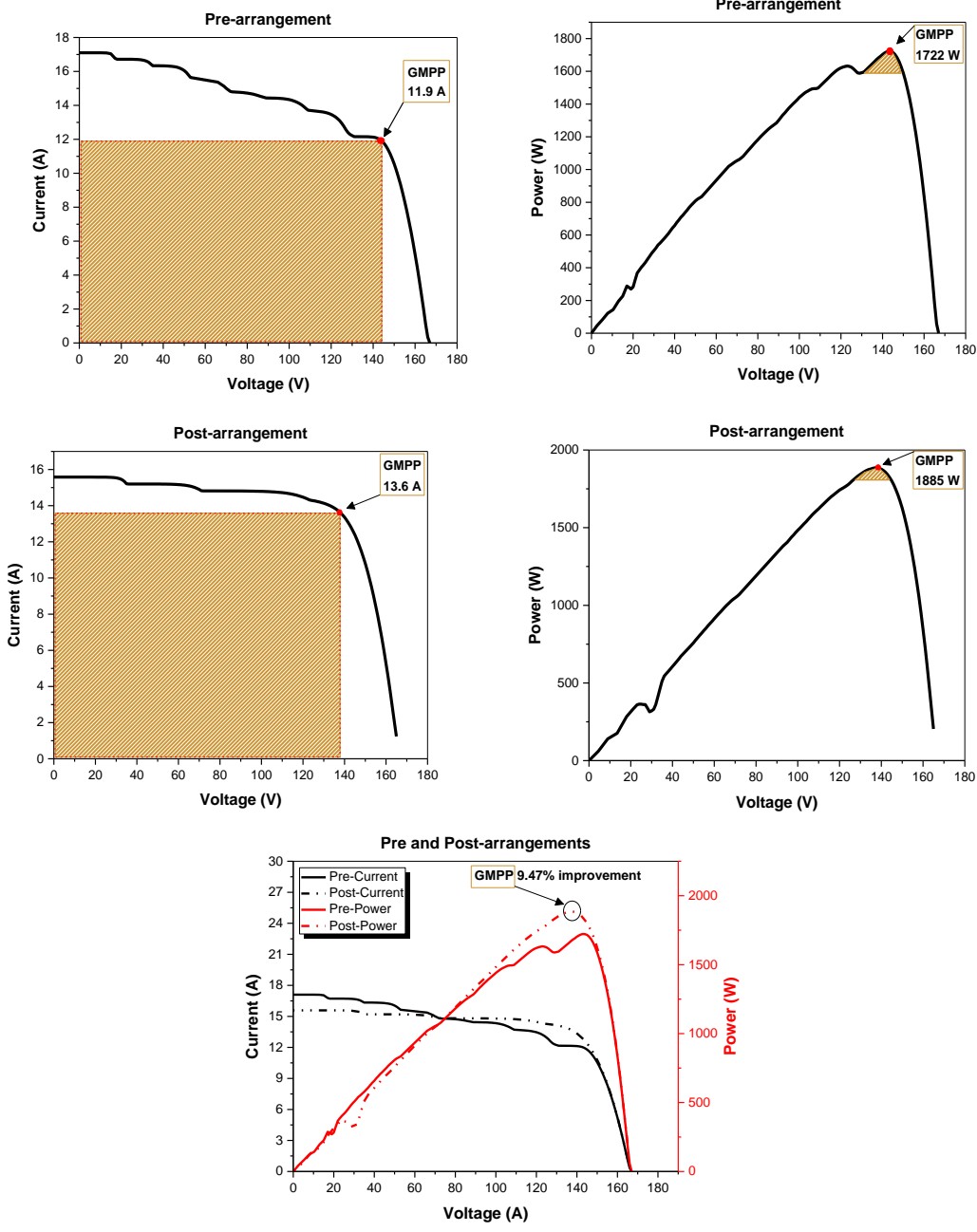

**Figure 12.** The output of the Array (pre-post rearrangements) for case 2.

**Table 7.** PV array configuration for case 3.

| Pre-Arrangement | | | | | | | |
|---|---|---|---|---|---|---|---|
| 0.4 pu | 0.6 pu | 0.4 pu | 0.6 pu | 0.9 pu | 0.6 pu | 0.8 pu | 0.5 pu |
| 0.8 pu | 0.4 pu | 0.9 pu | 0.9 pu | 0.8 pu | 0.5 pu | 0.6 pu | 0.6 pu |
| 0.6 pu | 0.8 pu | 0.7 pu | 0.5 pu | 0.6 pu | 0.8 pu | 0.5 pu | 0.8 pu |
| 0.6 pu | 0.8 pu | 0.7 pu | 0.5 pu | 0.6 pu | 0.8 pu | 0.6 pu | 0.4 pu |
| 0.4 pu | 0.4 pu | 0.9 pu | 0.4 pu | 0.6 pu | 0.6 pu | 0.5 pu | 0.4 pu |
| 0.7 pu | 0.8 pu | 0.9 pu | 0.5 pu | 0.5 pu | 0.7 pu | 0.4 pu | 0.5 pu |
| 0.5 pu | 0.7 pu | 0.4 pu | 0.9 pu | 0.9 pu | 0.6 pu | 0.9 pu | 0.7 pu |

**Table 7.** *Cont.*

| Post-Arrangement | | | | | | | |
|---|---|---|---|---|---|---|---|
| 0.9 pu | 0.9 pu | 0.9 pu | 0.9 pu | 0.9 pu | 0.9 pu | 0.9 pu | 0.9 pu |
| 0.8 pu | 0.8 pu | 0.8 pu | 0.8 pu | 0.8 pu | 0.8 pu | 0.8 pu | 0.8 pu |
| 0.7 pu | 0.7 pu | 0.7 pu | 0.7 pu | 0.6 pu | 0.6 pu | 0.7 pu | 0.6 pu |
| 0.6 pu | 0.6 pu | 0.6 pu | 0.6 pu | 0.6 pu | 0.6 pu | 0.6 pu | 0.6 pu |
| 0.5 pu | 0.5 pu | 0.5 pu | 0.5 pu | 0.5 pu | 0.5 pu | 0.5 pu | 0.5 pu |
| 0.5 pu | 0.5 pu | 0.4 pu | 0.5 pu | 0.5 pu | 0.4 pu | 0.4 pu | 0.5 pu |
| 0.4 pu | 0.4 pu | 0.4 pu | 0.4 pu | 0.4 pu | 0.4 pu | 0.4 pu | 0.4 pu |

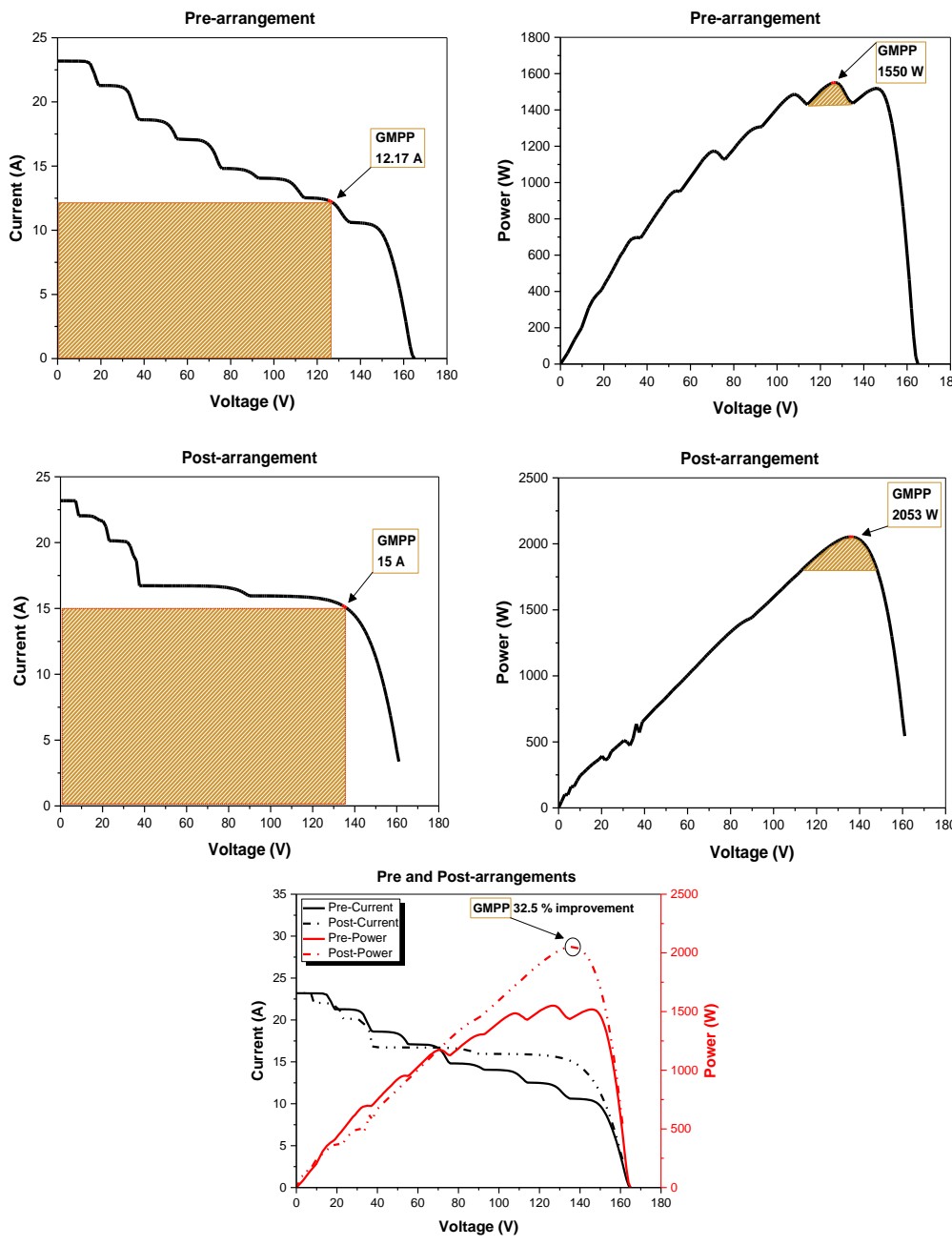

**Figure 13.** The output of a PV array (pre-post rearrangements) for case 3.

**Table 8.** PV array $7 \times 8$ parameters Pre and Post-arrangement.

| PV Array $7 \times 8$ Parameters | | | | |
|---|---|---|---|---|
| Parameters | Pre-Arrangement | Post-Arrangement | Power Improvement | Computing Time (s) |
| Current $I_{mpp}$ | 12.17 A | 15 A | – | – |
| Voltage $V_{mpp}$ | 127 V | 136 V | 32.5% | 5.64 |
| Power $W_L$ | 1550 W | 2053 W | – | – |

## 6. Restriction of Inverter Voltage

The restrictions of lowest and highest inverter voltage have to be taken into account in each scenario because electricity provision by a PV array to ac users cannot occur without an inverter [7]. Thus, prospective chains with inadequate inverter voltage will neither be accepted nor verified.

## 7. Discussion

The applicability of the suggested algorithm to different PV array dimensions and its ability to enhance maximum power output for every dimension considered is proven by the results obtained. By repositioning individual PV modules in every string based on their suitable AFs, the algorithm can also attenuate the effect of the bypass diodes, thus reducing the implications of incompatibility losses across PV modules in a particular string. On the downside, attention was not paid to voltage drawbacks, although these have been addressed in [3]. PV modules are sorted by the suggested algorithm hierarchically and repetitively. The post-arrangement minimisation of the impact of incompatibility among PV modules is indicated by the output resulting in P-V curves for the three scenarios investigated Figure 14. The capability of the algorithm to speedily generate results stems from the fact that it does not need to access all potential configurations for a given PV array. To give an example, the ideal PV module configuration was determined by the algorithm in the first scenario based solely on five steps and, the potential 2,627,625 arrangements did not have to examine in their entirety. Tables 4, 6 and 8 respectively show the computational times for each scenario. Thus, it is apparent that the ideal module arrangement can be identified quickly by the suggested algorithm and subsequently applied rapidly in real-time. Moreover, the algorithm is useful because it only repositions the damaged PV modules, while the others are left unchanged; thereby reducing the number of relays necessary for switching purposes. Makes the algorithm more cost-effective and less time-consuming than other strategies [3,7,22,29]. Because of the electric switches to achieve online reconfiguration that would need a large number of switches and a large number of cables, the high cost makes this kind of solution unaffordable in a real application.

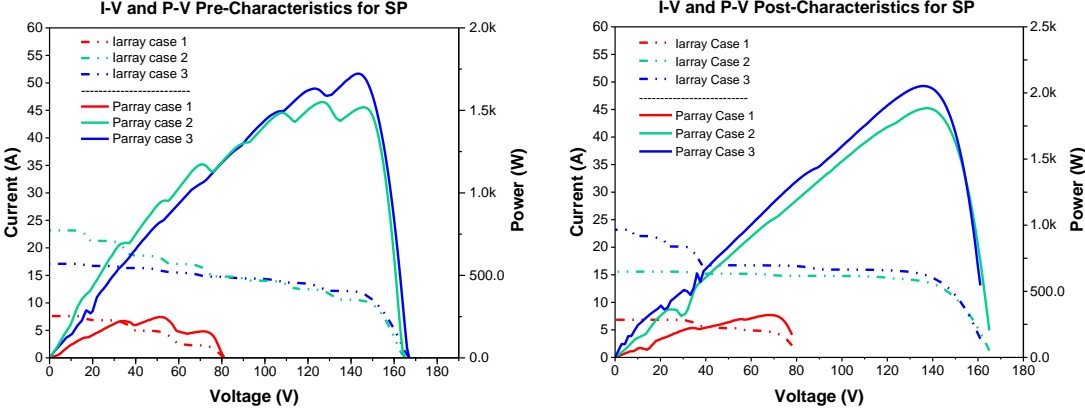

**Figure 14.** The outputs of a PV Arrays pre and post reconfigurations.

## 8. Conclusions

Non-uniform aging processes in PV arrays are the focus of the present paper, with results showing that the power production by of these arrays is affected by the positions of aged PV modules in the PV arrays. A new algorithm for rearranging PV arrays is therefore put forward to attenuate the impact of PV arrays with non-uniform aging and to increase the amount of power that they can produce while precluding the necessity to substitute the aged PV modules. Furthermore, to minimise the incompatibility effect caused by the non-uniform aging between PV modules, the algorithm sorted the PV modules in a repetitive and hierarchical manner. Thus, the maximum power output was increased by 29.7% for the 3 × 4 PV array, by 9.47% for the 5 × 8 array, and by 32.5% for the 7 × 8 array. It can be concluded that the suggested strategy for reconfiguring PV modules can successfully increase the maximum power output of PV systems with a lower number of relays than the current online approaches for the rearrangement of PV arrays. Wherefore, the plan for reconfiguration depends on the cost and benefit. So, providing the aging map of a PV plant is requisite, which propose a reconfiguration method to calculate the efficiency improvement and the corresponding profit; and then the workforce cost for reconfiguration its needs to be calculated. Consequently, If the profit in more power generation can cover the cost of the workforce in the reconfiguration, then it is suggested that the PV plant owner do undergoes reconfiguration to improve the benefits. Therefore, the advantage of the proposed strategy is to employ a workforce to swap PV modules' positions only.

**Author Contributions:** Conceptualization, M.A. and K.N.; Methodology, M.A.; Software, M.A. and Z.W.; Validation, K.N. and C.S.K.; Formal analysis, C.S.K., M.A.; Investigation, M.S.A. and C.S.K.; Resources, M.A.; Data curation, M.S.A.; Writing—original draft preparation, M.A.; Writing—review and editing, Z.W. and M.S.A.; Visualization, Z.W.; Supervision, K.N. and C.S.K.; Project administration, K.N. and C.S.K. All authors have read and agreed to the published version of the manuscript.

**Funding:** This research received no external funding.

**Conflicts of Interest:** The authors declare no conflict of interest.

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
