# Peer review of "A Novel PV Array Reconfiguration Algorithm Approach to Optimising Power Generation across Non-Uniformly Aged PV Arrays by Merely Repositioning"

_2571-8800, doi:10.3390/j3010005_

Round 1

Reviewer 1 Report

The authors propose in their work an optimized algorithm for the reconfiguration of PV panels in order to overcome the loss of energy in PV arrays formed by non-uniformly aged PV modules. This is not a new idea, but the approach presented by the Authors would be interesting if it is presented adequately.

In my opinion, some of the topics presented are not adequately described and they must be corrected before your paper become suitable for publication.

Minor mistakes:

1) In equations 2, 3, 4, 5 and 6 some mathematical operators do not appear (they have been replaced by a square).

2) Something in equation 8 is incorrect please review it.

3) In lines 159 and 160, the authors must use the term “PV module” instead the term “PV cell”.

4) In line 162 the use of short-circuit current is a mistake. The correct parameter is the maximum power current.

5) The acronym “pu” is used in figures 6 (line 182) and 10 (line 230) but it is not defined as “per unit” until line 245. The acronym must be defined the first time used in document.

6) The equation embedded in line 234 is incomprehensible. It must be explained and correctly rewrite.

7) According with the definition presented in line 220, (NxM PV array where N is the number of strings in parallel and M is the number of PV modules in series), the names used for the PV arrays considered by authors are wrong. The arrays simulated are 3x4, 5x8 and 7x8 PV arrays. Please review the nomenclature used.

8) The reviewer disagree with the Authors’ affirmation introduced in line 172: “The modules must be the same for equation (7) and equation (8) to be applicable”. These two equations (in the correct form) are derived from the application of the Kirchhoff’s laws and, as a consequence, they can be applied to any number and type of PV modules associated in series, even if the PV modules are not equal.

Points to consider:

1) The equivalent circuit of PV array presented in figure 4 is incorrectly adapted from reference [25]. The values of the equivalents Rs and Rsh are wrong; please use the correct expression presented in your reference [25] or in the original research paper: Huan-Liang Tsai et al. “Development of Generalized Photovoltaic Model Using MATLAB/SIMULINK” 2008.

In any case, if the authors have used the incorrect expression in the models utilized in the conducted simulations, they must be repeated with the correct model.

2) In lines 164 and 165 authors claim: “The short-circuit current fluctuates more widely compared to the open-circuit voltage caused by the p-n junction features of the PV cell as the latter ages according to the author [28]”, and I disagree with this consideration.

In one hand, I do not find nothing similar to this assertion in reference [28] and, in the other hand, it is possible find references along the time that shows the effect of aging over Rs and, consequently, over the open-circuit voltage. Here some examples:

- Amina Azizi et al. “Impact of the aging of a photovoltaic module on the performance of a grid-connected system” 2018.

- Bechara Nehme et al. “Contribution to the modeling of ageing effects in PV cells and modules” 2014.

- Kaplani, E. “Ageing effects in PV cells and modules” 2012.

3) The reviewer do not understand the paragraph “3. Disclosure of PV Aging”. The TDR method allows the identification a of PV module performance but, in what way is their use proposed by authors? Is it used for the aging coefficient determination? In any case, Authors must clearly specify what the proposed procedure for the aging coefficient determination is.

4) If Authors evaluate the PV modules aging using the short-circuit current (as is indicated in lines 165 and 166), this procedure has a serious drawback in case of real-time applications. During the time used for the short-circuit current measurement, the PV module must be isolated from the PV array and, as a direct consequence, the energy production of the PV array is drastically reduced. Authors must be more specific indicating how the aging coefficient is calculated o determined and the consequences of this procedure in the PV array energy production.

5) The PV short-circuit current (and the aging coefficient in this work) is directly related with the incident solar irradiance. The equalization by rows of the aging coefficient looks similar to the equalization by rows of the solar irradiance, and this last procedure was introduced several years ago in this reference: Velasco, G. et al. “Electrical PV Array Reconfiguration Strategy for Energy Extraction Improvement in Grid-Connected PV Systems” 2009. An extensive revision of the available literature about PV arrays reconfiguration based on irradiance equalization is suggested in order to improve the contextualization of the work presented by Authors.

6) The reviewer disagree with Authors in the opinion expressed in line 248: “… the optimization issue is addressed in the present work based on a genetic algorithm”. In my opinion, the algorithm proposed by Authors is a kind of iterative sorting algorithm but not based in the principles of the genetic algorithms (a method for solving both constrained and unconstrained optimization problems that is based on natural selection).

https://towardsdatascience.com/introduction-to-genetic-algorithms-including-example-code-e396e98d8bf3

7) The document presented does not address one of the main drawbacks of the reconfigurable PV arrays: the switch set. In this regard, I think that if the authors could present some information related with the switch set practical implementation, the interest of this paper would be greater. Therefore, Authors could try to clarify the following doubtful statement: “Moreover, the algorithm is useful because it only repositions the damaged PV modules, while the others are left unchanged, thereby minimizing the number of relays necessary for switching”.

8) Finally, I have two last questions: How many PV array reconfigurations are expected in a year due to the uneven aging of the PV modules? Is the use of a reconfiguration matrix in these circumstances justifiable?

In conclusion, and in my humble opinion, this paper should be submitted to a thorough revision.

Yours sincerely,

Author Response

Thank you very much for your precious time and great effort in reviewing our manuscript. We have made the corresponding modifications according to your valuable comments.

Regards,

Kai

Reviewer 2 Report

Alkahtan et al. performed a researched on energy potential of a non-uniform PV arrays. I suggest following revisions:

1- Introduction can be improved by introducing recent publications in the area of PV especially related applications. I suggest one below and you may add more:

https://www.sciencedirect.com/science/article/pii/S0038092X19302609

2- A comparison may be done with other similar research as a verification/validation.

3- How can the proposed model or approach be extended to include more practical situation or more generalized cases.

4- How did you consider the effect of inverters in you research? 

Author Response

(The authors gave the same response as above.)

Round 2

Reviewer 1 Report

Dear Colleagues,

You have satisfactorily considered some of the comments made by the reviewers. However, and in my humble opinion, you still have to clarify some subjects or issues of your work.

1) Please, pay attention to the definition "N x M PV array" and the use made of it in the document:

Lines 218-219 and Figure 8 caption define “N” as the number of parallel-connected strings and “M” as the number of series-connected PV modules.

Nevertheless, in lines 226-227 and Figure 9 caption use “N” as the number of series-connected PV modules an “M” as the number of parallel-connected strings.

In concordance with the definition, the PV array shown in figure 9 is a “3 x 4 PV array” and not a “4 x 3 PV array”. This mistake is present in the entire document.

2) The affirmation presented in line 176: “The modules must be the same for equation (7) and equation (8) to be applicable” is not correct. These two equations derive from the Kirchhoff’s laws and consequently, they are applicable to any number and type of PV modules associated in series and/or in parallel, even if the PV modules are not the same. I consider that this phrase must be clarified or deleted.

3) The equivalent circuit of PV array presented in figure 4 is incorrectly adapted from references: The values of the equivalents Rs and Rsh are wrong. Please review this model and use the correct expression.

4) The reviewer continues to disagree with Authors’ opinion expressed in lines 246 to 248. The algorithm proposed by Authors is a kind of iterative sorting algorithm but it is not based in the principles of genetic algorithms. Please, clarify your assertion or delete this part of the sentence.

5) The document presented does not address one of the main drawbacks of the reconfigurable PV arrays: the switching strategy implementation. In the answers to Reviewers Authors say:

“The background of this paper is to employ workers to swap PV modules”

“The plan for reconfiguration depends on the cost and benefit. For example, when we get the aging map of a PV plant, we will use the proposed reconfiguration method to calculate the efficiency improvement and the corresponding profit; and then we need to calculate the worker cost for reconfiguration. If the profit in more power generation can cover the cost of workers in the reconfiguration, we would like to suggest the PV plant owner to do reconfiguration to improve the benefit”.

These two ideas are very important (almost fundamental) for the Authors’ work understanding and must be included in the document.

6) Finally, I think your paper needs include some considerations about the practical implementation of your proposals:

How do the Authors propose to obtain the aging map of the PV plant? Do they propose to carry out the measures through a data acquisition system or through workers? How often do you intend to perform the measurement of the PV plant aging map? Does measuring of the PV plant aging map mean stopping the plant? What can be the consequences?

In my opinion, your work still needs a thorough new review.

Yours sincerely,

Author Response

(The authors gave the same response as above.)

Round 3

Reviewer 1 Report

Dear Colleagues,

You have successfully considered the comments made by the reviewers and in my opinion, this article may be published. Congratulations for your interesting work.

Dr. Guillermo Velasco

Associate Professor

Universitat Politècnica de Catalunya – Barcelona TECH